# Towards Redundancy Reduction in Diffusion Models for Efficient Video Super-Resolution

## Abstract

Diffusion models have recently shown promising results for video super-resolution (VSR). However, directly adapting generative diffusion models to VSR can result in redundancy, since low-quality videos already preserve substantial content information. Such redundancy leads to increased computational overhead and learning burden, as the model performs superfluous operations and must learn to filter out irrelevant information. To address this problem, we propose OASIS, an efficient **o**ne-step diffusion model with **a**ttention **s**pecialization for real-world v**i**deo **s**uper-resolution. OASIS incorporates an attention specialization routing that assigns attention heads to different patterns according to their intrinsic behaviors. This routing mitigates redundancy while effectively preserving pretrained knowledge, allowing diffusion models to better adapt to VSR and achieve stronger performance. Moreover, we propose a simple yet effective progressive training strategy, where training starts with temporally consistent degradations and then shifts to inconsistent settings. This strategy facilitates learning under complex degradations. Extensive experiments demonstrate that OASIS achieves state-of-the-art performance on both synthetic and real-world datasets. OASIS also provides superior inference speed, offering a **6.2×** speedup over one-step diffusion baselines such as SeedVR2. The code and models will be publicly available.

## 1 Introduction

Video super-resolution (VSR) is a widely studied task that aims to reconstruct high-resolution videos from low-resolution inputs (Jo et al., 2018). With the explosive growth of social media, enhancing real-world videos has become increasingly important. In contrast to synthetic degradations, such as bicubic downsampling, real-world videos often undergo far more diverse and unpredictable degradations, including varying levels of blur, noise, and compression artifacts (Chan et al., 2021; Wang et al., 2021). These complex conditions substantially increase the difficulty of restoring accurate and temporally consistent high-resolution content, making real-world VSR a challenging problem.

Recent advancements of diffusion models (Ho et al., 2020; Song et al., 2020), particularly diffusion transformer (DiT) architectures(Peebles & Xie, 2023; Ma et al., 2024), have achieved remarkable success in both image and video generation (Rombach et al., 2022; Yang et al., 2024b; Wan et al., 2025), demonstrating strong spatiotemporal modeling capacity. This progress naturally motivates the adaptation of pretrained diffusion models to video restoration tasks (Zhou et al., 2024; Li et al., 2025; Du et al., 2025), especially under the one-step diffusion paradigm (Liu et al., 2025; Wang et al., 2025a), which accelerates inference while preserving generative power.

However, most diffusion-based VSR methods share a key limitation: they overlook the redundancy in pretrained diffusion models (Yuan et al., 2024) when adapted to VSR, which arises because low-quality videos already preserve content information. This potential redundancy complicates adaptation, as VSR models must learn to disentangle it. Rather than mitigating redundancy, existing works typically improve performance by adding extra modules such as ControlNet (He et al., 2024; Xie et al., 2025), temporal layers (Zhou et al., 2024; Li et al., 2025), or optical-flow networks (Yang et al., 2024a). Yet these methods lead to higher complexity but limited benefits. Other methods (Wang et al., 2025a;b) attempt architectural redesigns to reduce redundancy, but such changes disrupt pretrained knowledge, thus requiring expensive retraining (*e.g.*, 256 H100-80G GPUs). These drawbacks highlight the need for more efficient ways to adapt pretrained diffusion models for VSR.

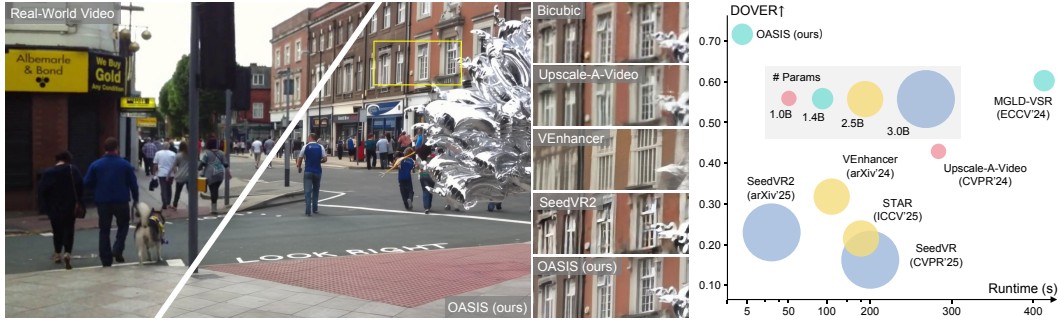

Figure 1: Inference speed and performance comparisons. The running time is evaluated on an A100 GPU using a 33-frame 720×1280 video, while DOVER is reported on the MVSR4x dataset. Our OASIS demonstrates superior reconstruction quality over existing diffusion-based methods, producing clearer and more faithful details. At the same time, OASIS also provides higher inference efficiency. Compared with SeedVR2 (Wang et al., 2025a), it runs approximately 6.2× faster.

To alleviate the aforementioned limitations, we investigate redundancy in existing diffusion-based video generative models. We find that one major redundancy lies in the attention mechanism in DiTs. While most diffusion models (Kondratyuk et al., 2023; Yang et al., 2024b; Kong et al., 2024; Wan et al., 2025) employ global attention uniformly for all attention heads, many heads consistently behave in a localized manner across different videos and scales. Specifically, apart from global attention, two dominant localized patterns emerge, including intra-frame attention and window attention, where the former mainly captures dependencies within a single frame, and the latter restricts interactions to local spatio-temporal windows. This indicates that diffusion transformers naturally specialize at the head level, with different heads focusing on global or local information. Thus, applying global attention across all heads inevitably introduces redundancy. This can result in increased computational overhead and learning burden, as the model spends resources on unused global dependencies and must learn to filter out irrelevant patterns.

Motivated by this observation, we propose OASIS, an efficient **o**ne-step diffusion model with **a**ttention **s**pecialization for real-world video super-resolution. The key component of OASIS is the attention specialization routing. Instead of treating all heads as identical global processors, we compute the KL-divergence between the original global attention distributions and localized alternatives. Attention heads that align more closely with localized patterns are reassigned to the corresponding modes, while the rest retain global attention. This attention specialization routing reduces unnecessary computation, better matches the intrinsic functionality of each head, and facilitates adaptation of pretrained diffusion transformers to real-world VSR. As shown in Fig. 1, OASIS delivers superior reconstruction and inference speed over existing methods.

Moreover, real-world degradations are typically highly diverse and can vary across frames, which further complicates the learning process. To address this, we introduce a simple yet effective progressive training strategy. Specifically, in the first stage, the model is trained with temporally consistent degradations, where all frames in a video share the same degradation type and severity. This helps the model learn fundamental restoration capability before handling more complex degradations. In the second stage, the training shifts to temporally inconsistent degradations, where each frame undergoes frame-wise varying distortions. This stage better reflects real-world conditions and encourages the model to handle frame-wise variations, thereby improving robustness.

Our main contributions are summarized as follows:

- We propose a novel and efficient one-step diffusion model, OASIS, for real-world VSR. By incorporating an attention specialization routing, OASIS mitigates the redundancy of pretrained diffusion transformers when adapted to VSR, thereby reducing computational cost and making the model better leverage diverse attention patterns for high-quality restoration.

- We design a simple yet effective progressive training strategy, where the model is first trained with temporally consistent degradations and then with temporally inconsistent settings to better reflect real-world scenarios. This strategy reduces the learning burden, enabling the model to handle complex degradations more effectively.

- OASIS achieves state-of-the-art results on multiple benchmarks, excelling in both quantitative metrics and perceptual quality. Moreover, it provides remarkable inference efficiency compared with existing diffusion-based VSR methods.

# 2 RELATED WORK

## 2.1 VIDEO SUPER-RESOLUTION

In recent years, learning-based approaches have driven significant progress in video super-resolution (VSR) (Isobe et al., 2020; Chan et al., 2021; 2022a; Li et al., 2023; Chen et al., 2024). These methods exploit diverse architectures, ranging from deformable convolutions (Wang et al., 2019; Tian et al., 2020) to transformer-based designs (Li et al., 2020; Liang et al., 2022; Shi et al., 2022). Inspired by the success of GANs in image restoration, several GAN-based methods (Lucas et al., 2019; Xu et al., 2025) have also been introduced to recover fine-grained details.

Despite these advances, existing methods often struggle under complex real-world degradations. To enhance robustness, some works (Yang et al., 2021; Wang et al., 2023b) leverage real-world LQ–HQ paired data to enhance robustness. Others focus on architectural redesigns (Pan et al., 2021; Wu et al., 2022; Zhang & Yao, 2024) to improve adaptability to challenging degradations. In parallel, various degradation pipelines have been proposed to better simulate real-world conditions (Wang et al., 2021; Chan et al., 2022b). Nevertheless, these methods still exhibit limited performance when faced with diverse and unpredictable real-world degradations.

## 2.2 DIFFUSION MODELS

Diffusion models are powerful generative frameworks that synthesize structured data from random noise via iterative denoising (Ho et al., 2020; Song et al., 2020). Recently, they have achieved strong performance in both image (Rombach et al., 2022; Podell et al., 2023) and video (Ho et al., 2022; Zheng et al., 2024; Yang et al., 2024b; Wan et al., 2025) generation. However, multi-step diffusion models are often hindered by slow inference, motivating the development of one-step approaches for acceleration (Liu et al., 2022; Song et al., 2023; Yin et al., 2024; Lin et al., 2025)

Owing to the strong generative prior, diffusion models have also shown competitive performance in image and video restoration (Zhou et al., 2024; Guo et al., 2025a;b; Li et al., 2025; Wang et al., 2025b). For instance, Upscale-A-Video (Zhou et al., 2024) extends image diffusion models with temporal layers for video sequences, while MGLD-VSR (Yang et al., 2024a) leverages optical flow to refine latent sampling for better temporal coherence. STAR (Xie et al., 2025) incorporates a local enhancement module to restore fine details, and SeedVR (Wang et al., 2025b) employs a sliding-window strategy to handle long sequences. More recently, several works have explored one-step acceleration for faster inference (*e.g.*, SeedVR2 (Wang et al., 2025a); (Liu et al., 2025; Sun et al., 2025)). However, most existing approaches overlook the inherent redundancy in pretrained diffusion models, which limits their effectiveness when directly adapted to VSR.

## 2.3 REDUNDANCY REDUCTION IN LATENT DIFFUSION MODELS

Recent studies have highlighted that diffusion models, despite their strong generative power, often suffer from substantial redundancy (Sun et al., 2024b; Zhang et al., 2023; 2025a; Zhao et al., 2024), which becomes especially pronounced in restoration tasks (Chen et al., 2025a) since low-quality videos already contain much of the underlying content. To address this, a growing body of work has focused on reducing redundancy to improve efficiency (Castells et al., 2024; Zhu et al., 2023; Zhang et al., 2024; Pu et al., 2024; Sun et al., 2024a; Tian et al., 2025; Fang et al., 2025).

In particular, attention redundancy in diffusion transformers has attracted considerable attention. DiTFastAttn (Yuan et al., 2024) reduces redundant computation through attention sharing and localized attention patterns. SVG (Xi et al., 2025) leverages the inherent sparsity of 3D spatiotemporal attention by profiling head types and applying sparse patterns with kernel optimizations, while STA (Zhang et al., 2025b) eliminates redundancy from global attention with hardware-aware sliding window design. Nevertheless, how to reduce redundancy in diffusion-based VSR methods while simultaneously improving performance remains unexplored.

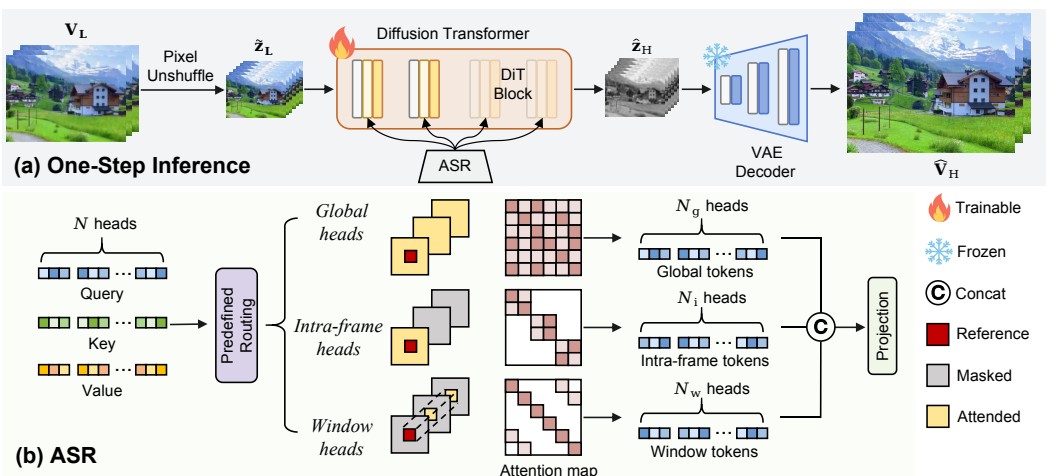

Figure 2: Overview of OASIS. Given an input LQ video, a pixel-unshuffle operation maps it into the latent space, which is then processed by a diffusion transformer with attention specialization routing (ASR). ASR reduces redundancy by dividing attention heads into global, intra-frame, and window groups to capture complementary contexts. Their outputs are concatenated into an aggregated feature, and a VAE decoder reconstructs the HQ video from the restored latent.

## 3 METHOD

### 3.1 PRELIMINARIES: ONE-STEP LATENT DIFFUSION MODEL

Latent Diffusion Models (Rombach et al., 2022) are formulated in a low-dimensional latent space, leading to improved efficiency in training and sampling. In the forward process, a clean latent $\mathbf{z}_0$ is gradually perturbed by Gaussian noise. At timestep $t$, the corrupted latent can be written as:

$$\mathbf{z}_t = \alpha_t \mathbf{z}_0 + \sigma_t \boldsymbol{\epsilon}, \quad \boldsymbol{\epsilon} \sim \mathcal{N}(\mathbf{0}, \mathbf{I}), \tag{1}$$

where $\alpha_t$ and $\sigma_t$ are the noise schedule conditioned on the timestep. The reverse process then reconstructs the clean latent through a learned prediction model (Ho et al., 2020), where transformer-based architectures (Peebles & Xie, 2023; Ma et al., 2024) have demonstrated strong performance. In the one-step setting, the network $\boldsymbol{\epsilon}_\theta$ directly estimates the clean latent $\hat{\mathbf{z}}_0$ from the noised latent:

$$\hat{\mathbf{z}}_0 = (\mathbf{z}_t - \sigma_t \boldsymbol{\epsilon}_\theta(\mathbf{z}_t, t)) / \alpha_t. \tag{2}$$

### 3.2 OVERVIEW OF OASIS

An overview of OASIS is shown in Fig. 2, which is built upon Wan2.1 (Wan et al., 2025), a powerful pretrained text-to-video diffusion model. Following prior work (Chen et al., 2025a), we omit the VAE encoder to avoid redundant encoding and instead apply a pixel-unshuffle operation (Shi et al., 2016) followed by a linear projection to directly map the input LQ video $\mathbf{V}_L$ into the latent space:

$$\tilde{\mathbf{z}}_L = \text{UnShuffle}(\mathbf{V}_L), \quad \mathbf{z}_L = \mathbf{W}_{\text{proj}}\tilde{\mathbf{z}}_L + \mathbf{b}_{\text{proj}}, \tag{3}$$

where $\mathbf{W}_{\text{proj}}$ and $\mathbf{b}_{\text{proj}}$ are the parameters of the linear projection layer. Unlike standard diffusion models that start from Gaussian noise, OASIS views LQ latents $\mathbf{z}_L$ as intermediate diffusion states and their high-quality (HQ) counterparts $\mathbf{z}_H$ as the clean target state. Since Wan2.1 adopts the flow matching formulation, the reconstruction can thus be formulated as:

$$\hat{\mathbf{z}}_H = \mathbf{z}_L - \sigma_{T_L}\mathcal{DN}_\theta(\mathbf{z}_L, T_L), \tag{4}$$

where $\hat{\mathbf{z}}_H$ is the estimated high-quality video latents, $\mathcal{DN}_\theta$ is the DiT integrated with our proposed attention specialization routing (ASR), and $T_L$ is the predefined timestep. Finally, the reconstructed HQ video $\hat{\mathbf{V}}_H$ is decoded from $\hat{\mathbf{z}}_H$ using the 3D VAE decoder $\mathcal{D}_\phi$: $\hat{\mathbf{V}}_H = \mathcal{D}_\phi(\hat{\mathbf{z}}_H)$.

### 3.3 REDUCING REDUNDANCY IN VIDEO DIFFUSION MODELS

#### 3.3.1 ATTENTION REDUNDANCY IN DIFFUSION TRANSFORMERS

When video generative diffusion models are adapted to VSR, they often introduce redundancy, as low-quality videos already retain content information. To address this, we investigate redundancy patterns in video generative diffusion models and find that attention is one common source.

As illustrated in Fig. 3, attention heads in DiTs exhibit distinct specialization patterns, which remain consistent for the same head across different videos. Some heads display global patterns (left), distributing attention across the entire sequence. In contrast, others focus on intra-frame patterns (middle), where attention mainly concentrates within each frame. Likewise, a subset of heads exhibits window patterns (right), where attention is concentrated in localized spatial neighborhoods. These visualizations highlight that, although formulated as global attention, not all heads truly exploit global context, revealing clear redundancy in uniform global attention assignments.

### 3.3.2 ATTENTION SPECIALIZATION ROUTING

The core idea of this routing strategy is to selectively replace global attention heads with localized alternatives, based on the similarities of their attention heatmaps. To this end, we first present the detailed formulations of two attention mechanisms, intra-frame and window attention, which we identify as the key localized patterns.

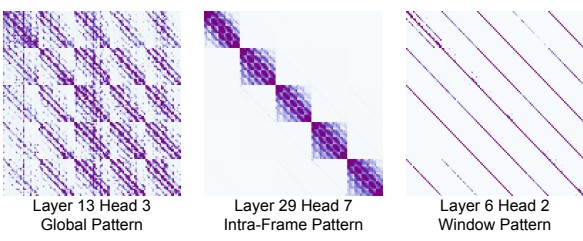

Layer 13 Head 3    Layer 29 Head 7    Layer 6 Head 2
Global Pattern    Intra-Frame Pattern    Window Pattern

Figure 3: Head-level specialization in diffusion transformers. Each visualization is the mean attention map over 50 videos from the training set, revealing global, intra-frame, and window patterns.

**Intra-Frame Attention.** Let the latents of the input LQ video be denoted as $\mathbf{z}_\mathrm{L} \in \mathbb{R}^{C \times T \times H \times W}$, where $C$ is the latent dimension, $T$ is the temporal length (number of frames), and $H, W$ are the spatial height and width, respectively. In diffusion transformers (DiTs), the query, key, and value tokens are obtained by applying separate learned linear projections to $\mathbf{z}_\mathrm{L}$. For each query token $q_{t,h,w} \in \mathbb{R}^d$ extracted from spatial position $(h, w)$ in frame $t$, where $d$ denotes the feature dimension of each head, the key and value sets are constructed by gathering all tokens within the same frame $t$. The intra-frame attention output is thus computed as:

$$\mathrm{Attn}_{\mathrm{intra}}(\mathbf{q}_{t,h,w}) = \sum_{h'=1}^{H} \sum_{w'=1}^{W} \frac{\exp\left(\mathbf{q}_{t,h,w}^\top \mathbf{k}_{t,h',w'}/\sqrt{d}\right) \mathbf{v}_{t,h',w'}}{\sum_{h'',w''} \exp\left(\mathbf{q}_{t,h,w}^\top \mathbf{k}_{t,h'',w''}/\sqrt{d}\right)}. \tag{5}$$

**Window Attention.** For each query token $\mathbf{q}_{t,h,w} \in \mathbb{R}^d$ at position $(t, h, w)$, the attention is restricted to a local spatiotemporal neighborhood. Specifically, we define a window of size $(P_t, P_h, P_w)$ centered at $(t, h, w)$, and gather all tokens whose indices fall within this window:

$$\mathcal{N}(t, h, w) = \left\{ (t', h', w') \,\middle|\, |t' - t| \le P_t/2, \; |h' - h| \le P_h/2, \; |w' - w| \le P_w/2 \right\}. \tag{6}$$

The window attention output is then computed as:

$$\mathrm{Attn}_{\mathrm{win}}(\mathbf{q}_{t,h,w}) = \frac{\sum_{(t',h',w')\in\mathcal{N}(t,h,w)} \exp\left(\mathbf{q}_{t,h,w}^\top \mathbf{k}_{t',h',w'}/\sqrt{d}\right) \mathbf{v}_{t',h',w'}}{\sum_{(t'',h'',w'')\in\mathcal{N}(t,h,w)} \exp\left(\mathbf{q}_{t,h,w}^\top \mathbf{k}_{t'',h'',w''}/\sqrt{d}\right)}. \tag{7}$$

To better exploit head-level specialization, we propose a simple algorithm to perform the routing. As illustrated in Algorithm 1, given a target ratio $\rho$ that specifies the proportion of attention heads to be retained as global, we evaluate each head on a set of videos by measuring the KL-divergence between its global attention heatmap and those from intra-frame and window attention. The smaller divergence value is taken as the head score $s_h$. Heads are then ranked by $s_h$ and sequentially replaced with the corresponding localized alternative until the remaining global heads meet the ratio $\rho$.

---

**Algorithm 1** Attention Specialization Routing

1: **Input:** DiT $\mathcal{DN}_\theta$, global-head ratio $\rho \in [0, 1]$
2: **Output:** Assignment map $\mathcal{A}$ (head → {global, intra-frame, window})
3: Initialize $\mathcal{A}(h) \leftarrow$ global for all heads;    $N \leftarrow$ number of heads in all layers
4: **For each head $h$ across all layers:**
5:      obtain attention map $M_h^g$, $M_h^i$, $M_h^w$ with global, intra-frame, and window attention
6:      $s_h^i \leftarrow \mathbb{E}\left[\mathrm{KL}(M_h^g \| M_h^i)\right]$;    $s_h^w \leftarrow \mathbb{E}\left[\mathrm{KL}(M_h^g \| M_h^w)\right]$
7:      $s_h \leftarrow \min\{s_h^i, s_h^w\}$;    $m_h \leftarrow \arg\min_{m \in \{i,w\}} s_h^m$
8: Sort all heads by $s_h$ (ascending);    $K \leftarrow \lceil \rho N \rceil$
9: Set $\mathcal{A}(h_j) \leftarrow m_{h_j}$ for $j = 1, \ldots, N - K$
10: **return** $\mathcal{A}$

---

### 3.3.3 REDUNDANCY BEYOND ATTENTION

For video super-resolution, diffusion models typically incorporate a VAE encoder to map frames into the latent space and a prompt extractor for conditioning (Zhou et al., 2024). The VAE encoder downsamples videos into latent space with a large network, which we consider unnecessarily complex. Therefore, we replace it with an extremely lightweight pixel-unshuffle (Chen et al., 2025a) operation. Moreover, since the prompt extractor derives features from the LQ video without introducing additional information, we omit it as well and use an empty prompt for conditioning instead. Our ablation study confirms that removing these modules has no adverse impact on performance.

## 3.4 PROGRESSIVE TRAINING

### 3.4.1 TWO-STAGE CURRICULUM LEARNING

Real-world degradations often exhibit temporal inconsistency across frames. For instance, unstable camera movement may result in motion blur in specific frames while others remain unaffected. This temporal inconsistency in degradations undoubtedly increases the difficulty of learning robust super-resolution models. To address this, we propose a progressive training strategy that guides the model to learn restoration in a curriculum manner, moving from simpler to more complex degradations.

The progressive training consists of two stages. In the first stage, we employ the second-order degradation model (Wang et al., 2021) to generate synthetic training data. Specifically, we randomly apply Gaussian noise, blur, and compression artifacts (image and video) to HQ videos. To control the difficulty, all frames within a video are assigned the same type and severity of degradation, ensuring temporal consistency across the sequence.

In the second stage, degradations are made temporally inconsistent across frames. To avoid excessive discontinuities, degradations are generated sequentially across frames (Chan et al., 2022b), with the type and severity conditioned on the previous frame and stochastically perturbed with a predefined probability. This setting more faithfully reflects real-world scenarios and further improves the model's robustness to diverse frame-wise variations. Our ablation studies demonstrate that progressive training clearly outperforms direct training on temporally inconsistent degradations.

### 3.4.2 TRAINING OBJECTIVES

In our progressive training, the two stages share the same training objective. Apart from the latent reconstruction loss for the one-step diffusion model, we further introduce perceptual and temporal losses in pixel space to enhance both visual fidelity and temporal consistency.

**Latent Reconstruction Loss.** Unlike standard diffusion models that optimize a noise-prediction loss (Ho et al., 2020), OASIS employs a latent reconstruction objective, which is defined as an MSE loss between $\hat{\mathbf{z}}_H$ and its ground-truth counterpart $\mathbf{z}_H$ over a mini-batch of size $B$:

$$\mathcal{L}_{\text{latent}}\left(\hat{\mathbf{z}}_H, \mathbf{z}_H\right) = \mathcal{L}_2\left(\hat{\mathbf{z}}_H, \mathbf{z}_H\right) = \frac{1}{B} \sum_{i=1}^{B} \|\hat{\mathbf{z}}_H - \mathbf{z}_H\|^2. \tag{8}$$

**Perceptual Loss.** Although the latent reconstruction loss provides direct supervision to the DiT, the target latent $\mathbf{z}_H$ obtained from the VAE encoder typically deviates slightly from the true HQ latent representations. To remedy this, we introduce a perceptual loss in pixel space, combining MSE and LPIPS (Zhang et al., 2018) to balance accuracy and visual quality:

$$\mathcal{L}_{\text{per}}(\hat{\mathbf{V}}_H, \mathbf{V}_H) = \mathcal{L}_2(\hat{\mathbf{V}}_H, \mathbf{V}_H) + \mathcal{L}_{\text{LPIPS}}(\hat{\mathbf{V}}_H, \mathbf{V}_H). \tag{9}$$

**Temporal Loss.** Supervision in pixel space is frame-wise and lacks explicit enforcement of temporal consistency. To strengthen the coherence of the restored HQ video, we extract optical flow (Teed & Deng, 2020) from the ground-truth video and warp each predicted frame toward its neighboring frame. The temporal loss is defined as the MAE between the warped frame and its neighbor:

$$\mathcal{L}_{\text{warp}} = \sum_{i=1}^{M} \| \text{Warp}(\hat{\mathbf{V}}_H^i, \mathbf{O}_{\text{GT}}^{\text{bw},i}) - \hat{\mathbf{V}}_H^{i+1} \|_1 + \| \text{Warp}(\hat{\mathbf{V}}_H^i, \mathbf{O}_{\text{GT}}^{\text{fw},i}) - \hat{\mathbf{V}}_H^{i-1} \|_1, \tag{10}$$

where $M$ is the number of frames, $\mathbf{O}_{\text{GT}}^{\text{bw},i}$ and $\mathbf{O}_{\text{GT}}^{\text{fw},i}$ are the backward and forward optical flow derived from the ground-truth video. The overall training objectives can thereby be expressed as:

$$\mathcal{L} = \mathcal{L}_{\text{latent}} + \mathcal{L}_{\text{per}} + \lambda_{\text{warp}} \cdot \mathcal{L}_{\text{warp}}. \tag{11}$$

| Dataset | Metric | RealBasicVSR CVPR 2022 | Upscale-A-Video CVPR 2024 | MGLD-VSR ECCV 2024 | VEnhancer arXiv 2024 | STAR ICCV 2025 | SeedVR ICCV 2025 | SeedVR2 arXiv 2025 | OASIS (ours) 2025 |
|---|---|---|---|---|---|---|---|---|---|
| UDM10 | PSNR ↑ | 24.13 | 21.72 | 24.23 | 21.32 | 23.47 | 24.39 | 25.39 | 25.63 |
| | SSIM ↑ | 0.6801 | 0.5913 | 0.6957 | 0.6811 | 0.6804 | 0.7083 | 0.7564 | 0.7579 |
| | LPIPS ↓ | 0.3908 | 0.4116 | 0.3272 | 0.4344 | 0.4242 | 0.3417 | 0.2868 | 0.2452 |
| | CLIP-IQA ↑ | 0.3494 | 0.4697 | 0.4557 | 0.2852 | 0.2417 | 0.2869 | 0.2906 | 0.5510 |
| | DOVER ↑ | 0.7564 | 0.7291 | 0.7264 | 0.4576 | 0.4830 | 0.5493 | 0.5646 | 0.7863 |
| | $E^*_{warp}$ ↓ | 3.10 | 3.97 | 3.59 | 1.03 | 2.08 | 3.84 | 2.59 | 1.94 |
| SPMCS | PSNR ↑ | 22.17 | 18.81 | 22.39 | 18.58 | 21.24 | 21.73 | 22.36 | 22.75 |
| | SSIM ↑ | 0.5638 | 0.4113 | 0.5896 | 0.4850 | 0.5441 | 0.5803 | 0.6136 | 0.5904 |
| | LPIPS ↓ | 0.3662 | 0.4468 | 0.3263 | 0.5358 | 0.5257 | 0.3297 | 0.2905 | 0.2634 |
| | CLIP-IQA ↑ | 0.3513 | 0.5248 | 0.4348 | 0.3188 | 0.2646 | 0.3946 | 0.4086 | 0.4693 |
| | DOVER ↑ | 0.6753 | 0.7171 | 0.6754 | 0.4284 | 0.3204 | 0.6150 | 0.6251 | 0.7242 |
| | $E^*_{warp}$ ↓ | 1.88 | 4.22 | 1.68 | 1.19 | 1.01 | 1.83 | 1.24 | 1.15 |
| YouHQ40 | PSNR ↑ | 22.39 | 19.62 | 23.17 | 19.78 | 22.43 | 21.96 | 23.61 | 23.75 |
| | SSIM ↑ | 0.5895 | 0.4824 | 0.6194 | 0.5911 | 0.6276 | 0.5920 | 0.6771 | 0.6417 |
| | LPIPS ↓ | 0.4091 | 0.4268 | 0.3608 | 0.4742 | 0.4744 | 0.3466 | 0.2754 | 0.2608 |
| | CLIP-IQA ↑ | 0.3964 | 0.5258 | 0.4657 | 0.3309 | 0.2805 | 0.4123 | 0.3811 | 0.4817 |
| | DOVER ↑ | 0.8596 | 0.8596 | 0.8446 | 0.6957 | 0.5525 | 0.8618 | 0.8384 | 0.8700 |
| | $E^*_{warp}$ ↓ | 3.08 | 6.84 | 3.45 | 0.95 | 3.39 | 3.44 | 3.42 | 2.74 |
| RealVSR | PSNR ↑ | 22.00 | 20.74 | 22.08 | 15.75 | 17.43 | 20.44 | 20.20 | 21.14 |
| | SSIM ↑ | 0.7166 | 0.5681 | 0.6805 | 0.4002 | 0.5215 | 0.6792 | 0.6977 | 0.6212 |
| | LPIPS ↓ | 0.2036 | 0.4163 | 0.2241 | 0.3784 | 0.2943 | 0.2416 | 0.2197 | 0.2018 |
| | CLIP-IQA ↑ | 0.3538 | 0.2134 | 0.4109 | 0.3880 | 0.3641 | 0.2924 | 0.2887 | 0.4357 |
| | DOVER ↑ | 0.7384 | 0.3587 | 0.7354 | 0.7637 | 0.7051 | 0.6747 | 0.7209 | 0.7800 |
| | $E^*_{warp}$ ↓ | 4.72 | 1.00 | 3.03 | 5.15 | 9.88 | 3.62 | 4.77 | 2.63 |
| MVSR4x | PSNR ↑ | 21.80 | 22.35 | 22.58 | 20.50 | 22.42 | 22.16 | 21.72 | 22.66 |
| | SSIM ↑ | 0.7045 | 0.7327 | 0.7399 | 0.7117 | 0.7421 | 0.7407 | 0.7566 | 0.7428 |
| | LPIPS ↓ | 0.4235 | 0.4012 | 0.3486 | 0.4471 | 0.4311 | 0.4543 | 0.3667 | 0.3246 |
| | CLIP-IQA ↑ | 0.4118 | 0.3235 | 0.3738 | 0.3104 | 0.2674 | 0.2271 | 0.2243 | 0.5711 |
| | DOVER ↑ | 0.6846 | 0.4276 | 0.6062 | 0.3164 | 0.2137 | 0.1554 | 0.2219 | 0.7243 |
| | $E^*_{warp}$ ↓ | 1.69 | 0.66 | 1.51 | 0.62 | 0.61 | 2.28 | 1.33 | 0.87 |

Table 1: Quantitative comparison on synthetic and real-world datasets. The best and second best results are colored with red and blue. OASIS excels across multiple datasets and metrics.

## 4 EXPERIMENTS

### 4.1 EXPERIMENTAL SETTINGS

**Experimental Settings.** We train our model on the HQ-VSR dataset (Chen et al., 2025b), which contains 2,055 videos. To generate paired LQ-HQ data, we adopt the RealBasicVSR (Chan et al., 2022b) degradation model, but extend it to two stages with temporally consistent and inconsistent degradations. For evaluation, we follow prior works (Zhou et al., 2024; Yang et al., 2024a) on both synthetic (UDM10 (Tao et al., 2017), SPMCS (Yi et al., 2019), YouHQ40 (Zhou et al., 2024)) and real-world (RealVSR (Yang et al., 2021), MVSR4x (Wang et al., 2023b)) datasets. All experiments use a ×4 upscaling factor. The metrics include PSNR, SSIM (Wang et al., 2004), and LPIPS (Zhang et al., 2018) as reference metrics, and CLIP-IQA (Wang et al., 2023a), DOVER (Wu et al., 2023), and $E^*_{warp}$ (*i.e.*, $E_{warp}$ ($\times 10^{-3}$) (Lai et al., 2018)) as no-reference metrics.

**Implementation Details.** Our method is built on Wan2.1 (Wan et al., 2025) (1.42B parameters). We train the DiT with our progressive strategy while freezing all other components. Training is conducted on 8 NVIDIA A6000 GPUs using AdamW (Loshchilov & Hutter, 2017) ($\beta_1$=0.9, $\beta_2$=0.999). Input videos consist of 17 frames at 320×640 resolution, with a batch size of 8. OASIS is trained for 25,000 iterations per stage with a learning rate of $1 \times 10^{-4}$. We set loss weight $\lambda_{\text{warp}}$ to 0.1, predefined timestep $T_L$ to 799. and window attention size to $(3, 5, 5)$. For ASR, the global-head ratio $\rho$ is 0.4, with attention assignments derived from 50 training videos from the training set.

### 4.2 MAIN RESULTS

**Quantitative Results.** As shown in Tab. 1, OASIS achieves superior performance, achieving first or second place on 27 of 30 reported results. It delivers top scores in pixel-level (PSNR and SSIM) and perceptual (LPIPS) fidelity, maintains superiority on video quality metrics (CLIP-IQA and DOVER), and shows competitive temporal consistency ($E^*_{warp}$). These results highlight that OASIS provides the most outstanding and balanced overall performance.

**Qualitative Results.** Figure 4 compares OASIS with leading baselines on synthetic and real-world videos. While other methods can preserve coarse structures, they often suffer from oversmoothed textures, contour artifacts, blurred grids, or color shifts. In contrast, OASIS recovers realistic details while maintaining fidelity to the original video. We also provide the temporal consistency visualization in Fig. 5, where our method delivers strong temporal coherence, yielding smooth frame-to-frame transitions while preserving accurate details.

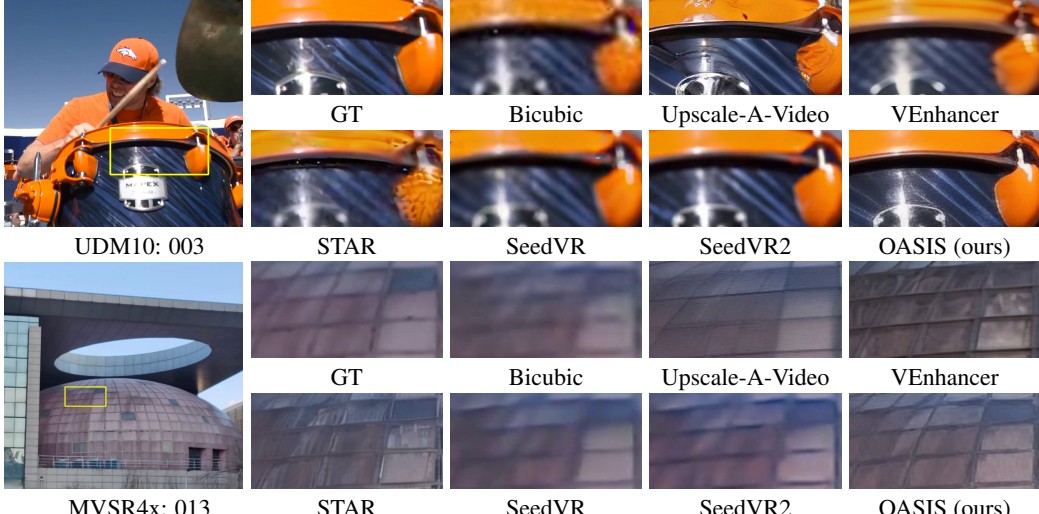

Figure 4: Visual comparisons on synthetic and real-world datasets for ×4 VSR. OASIS yields clean reconstructions while preserving contours and fine-scale surface patterns.

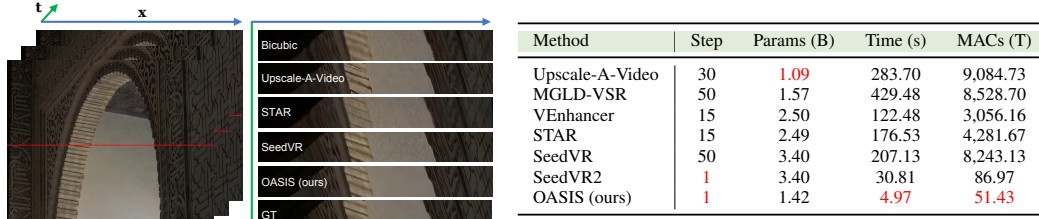

Figure 5: Comparison of temporal consistency. The temporal profile is obtained by stacking the red line across frames. Our method produces smoother frame transitions that closely resemble the ground truth.

Table 2: Comparison of inference steps (Step), number of parameters (Params), running time (Time), and multiply accumulate operations (MACs) of different diffusion-based VSR methods on a 33-frame 720×1280 video.

| Method | Step | Params (B) | Time (s) | MACs (T) |
|---|---|---|---|---|
| Upscale-A-Video | 30 | 1.09 | 283.70 | 9,084.73 |
| MGLD-VSR | 50 | 1.57 | 429.48 | 8,528.70 |
| VEnhancer | 15 | 2.50 | 122.48 | 3,056.16 |
| STAR | 15 | 2.49 | 176.53 | 4,281.67 |
| SeedVR | 50 | 3.40 | 207.13 | 8,243.13 |
| SeedVR2 | 1 | 3.40 | 30.81 | 86.97 |
| OASIS (ours) | 1 | 1.42 | 4.97 | 51.43 |

**Running Time Comparisons.** We evaluate efficiency in terms of inference steps, number of parameters, running time, and multiply accumulate operations (MACs) in Tab. 2. All methods are evaluated on one NVIDIA A100-80G GPU, generating a 33-frame 720×1280 video. Notably, OASIS achieves a significant reduction in computational cost, benefiting from its one-step diffusion design that accelerates inference and the attention specialization routine that mitigates redundancy.

## 4.3 ABLATION STUDY

This section studies the effect of each component in our method. All models are trained on the HQ-VSR dataset (Chen et al., 2025b) with a batch size of 4. For standard training, where the progressive strategy is not applied, each model is trained for 30,000 iterations. Under the progressive strategy, training is split into two stages, with 15,000 iterations performed for each stage.

**Attention Specialization Routine (ASR).** We compare ASR against models using only global, intra-frame, or window attention patterns. The global-head ratio $\rho$ is set to 0.4, and all models are trained using only stage 2 of the standard training configuration. As shown in Fig. 6, the hybrid design of ASR restores textures and details more faithfully than the single-pattern alternatives. In Tab. 3a, ASR consistently outperforms other variants across metrics. Moreover, compared with the original global-only design, ASR also provides higher efficiency. These results demonstrate the effectiveness of ASR in reducing redundancy while enhancing performance.

**Redundancy of VAE Encoder and Prompt Extractor.** We further study redundancy beyond attention, as shown in Tab. 3b. Following Upscale-A-Video, we adopt LLaVA (Liu et al., 2023) as the prompt extractor, with the prompt obtained from the first frame. Removing either module can improve efficiency. Since only the DiT is trained, the performance gains are less pronounced than ASR, but these results confirm the redundancy of these modules.

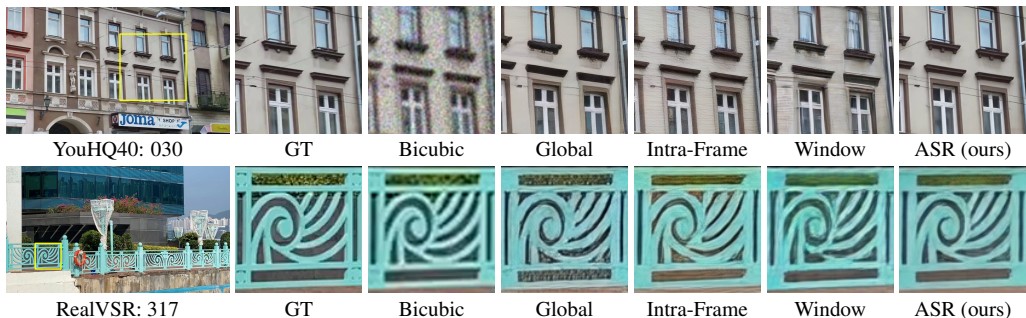

| YouHQ40: 030 | GT | Bicubic | Global | Intra-Frame | Window | ASR (ours) |
| RealVSR: 317 | GT | Bicubic | Global | Intra-Frame | Window | ASR (ours) |

Figure 6: Visual comparisons between ASR and different attention patterns.

| Method | PSNR↑ | LPIPS↓ | DOVER↑ | $E^*_{warp}$↓ | Time (s) |
|---|---|---|---|---|---|
| Global | 22.35 | 0.3418 | 0.7098 | 1.13 | 10.53 |
| Intra-Frame | 22.10 | 0.3324 | 0.6730 | 1.75 | 2.38 |
| Window | 22.28 | 0.3408 | 0.6831 | 1.18 | 6.60 |
| ASR (ours) | 22.77 | 0.3242 | 0.7142 | 0.87 | 7.50 |

(a) Ablation study of ASR.

| Method | PSNR↑ | LPIPS↓ | DOVER↑ | $E^*_{warp}$↓ | Time (s) | MACs (T) |
|---|---|---|---|---|---|---|
| w/ VAE+Prompt | 22.33 | 0.3433 | 0.6994 | 1.21 | 57.55 | 7.76 |
| w/o VAE | 22.31 | 0.3407 | 0.7007 | 1.16 | 55.79 | 6.03 |
| w/o Prompt | 22.41 | 0.3424 | 0.6901 | 1.17 | 53.09 | 6.84 |
| w/o VAE+Prompt | 22.35 | 0.3418 | 0.7098 | 1.13 | 51.43 | 4.97 |

(b) Ablation study of redundancy beyond attention.

| Method | PSNR↑ | SSIM↑ | LPIPS↓ | DOVER↑ | $E^*_{warp}$↓ |
|---|---|---|---|---|---|
| S1 | 22.23 | 0.7369 | 0.3378 | 0.6978 | 1.31 |
| S2 | 22.35 | 0.7336 | 0.3418 | 0.7098 | 1.13 |
| S1+S2 (ours) | 22.64 | 0.7432 | 0.3258 | 0.7144 | 0.97 |

(c) Ablation study of progressive training.

| $\mathcal{L}_{latent}$ | $\mathcal{L}_{per}$ | $\mathcal{L}_{warp}$ | PSNR↑ | SSIM↑ | LPIPS↓ | DOVER↑ | $E^*_{warp}$↓ |
|---|---|---|---|---|---|---|---|
| ✓ | | | 21.84 | 0.7744 | 0.4035 | 0.5871 | 2.07 |
| ✓ | ✓ | | 22.28 | 0.7351 | 0.3432 | 0.6896 | 1.54 |
| ✓ | ✓ | ✓ | 22.35 | 0.7336 | 0.3418 | 0.7098 | 1.13 |

(d) Ablation study of training loss functions.

Table 3: Quantitative results of the ablation study on the MVSR4x dataset. Best results are in red.

**Global-Head Ratio ($\rho$).** We evaluate ASR under different global-head ratios $\rho$ in Fig. 7. The ratios of each attention pattern under different values of $\rho$ are provided in the Appendix. At $\rho$=0.4, the model achieves the highest fidelity and temporal consistency, corresponding to the most effective specialization assignment. Notably, most values of $\rho$ outperform the global-only baseline, underscoring the soundness of the proposed attention specialization routing.

**Progressive Training Strategy.** We compare standard training against our progressive training in Tab. 3c. Training with stage 1 (S1) alone results in poor temporal consistency, while training with stage 2 (S2) alone also leads to suboptimal performance. In contrast, the progressive strategy (S1+S2) delivers clear improvements under the same number of training iterations. This highlights the effectiveness of progressive training in guiding the model from simple to complex degradations and enhancing robustness in real-world scenarios.

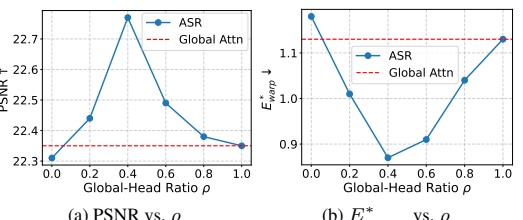

(a) PSNR vs. $\rho$

(b) $E^*_{warp}$ vs. $\rho$

Figure 7: Effect of global-head ratio $\rho$ on PSNR and $E^*_{warp}$ metrics. The results are evaluated on the MVSR4x dataset. Global Attn refers to the baseline using the global attention only.

**OASIS Training Loss Functions.** As shown in Tab. 3d, using only the latent reconstruction loss results in poor performance across all metrics, indicating that latent supervision alone is insufficient. In contrast, introducing the perceptual loss yields substantial improvements on all metrics, demonstrating the critical role of pixel-level supervision. Finally, adding the temporal loss further improves the $E^*_{warp}$ score, highlighting its importance in enhancing temporal consistency.

## 5 CONCLUSION

We propose OASIS, an efficient one-step diffusion model for real-world VSR. OASIS incorporates an attention specialization routine, which assigns DiT attention heads to global or localized patterns according to their attention distributions. This routine effectively reduces redundancy while improving performance. Moreover, we design a progressive strategy that trains the model from simple to complex degradations, enabling better adaptation to complex real-world scenarios. Extensive experiments highlight the superiority of OASIS over state-of-the-art methods with remarkable efficiency.

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
