# OpenReview forum: "Towards Redundancy Reduction in Diffusion Models for Efficient Video Super-Resolution"
_ICLR.cc/2026/Conference — ICLR 2026 Conference Withdrawn Submission_

### Official Review · Reviewer_KQQk · 2025-10-16

**Soundness:** 2
**Presentation:** 2
**Contribution:** 1
**Rating:** 2
**Confidence:** 5

**Summary:**

This paper proposes an efficient one-step diffusion model OASIS for real-world video super-resolution. It incorporates an attention specialization routine to assign DiT heads to one of global, intra-frame and window patterns. By introducing a two-stage curriculum learning strategy, it effectively adapts to complex real-world degradation pattern. Experiments demonstrate its superior efficiency and comparable performance.

**Strengths:**

* By seperating DiT heads into three patterns, it achieves notable efficiency in real-world scenario.
* It introduces a two-stage curriculum learning to effectively adapt T2V prior to Real-VSR settings.
* Detailed experiments demonstrate the effectiveness and efficiency of the proposed method.

**Weaknesses:**

* It is well-known that DiT heads can be seperated into intra-frame and inter-frame patterns. Also, previous methods like Sparse VideoGen have already reduced complexity by splitting the heads. What is the main difference between OASIS and these previous methods? Also, the seperation in OASIS is determined by a small set of videos in training set. However, authors did not provide experiments to show that the seperation is stable across different sets.
* The training set contains only 2k clips. However, it takes nearly 30k iterations to converge. A major concern is that the performance gain originates from over-fitting. Also, OASIS starts from a T2V prior, but removes its text encoder. From past experience, it usually leads to inferior visual quality and slow convergence speed. Authors need to provide more detailed explanation about it.
* As shown in previous methods like SeedVR, DiffVSR and PatchVSR, the second-order degradation pipeline proposed by RealBasicVSR is effective for real-world scenarios, and the VSR model can be directly trained in one-stage, which conflicts the conclusion in this paper. Authors need to provide more detailed evidence to demonstrate the validity of this conclusion. I am concerned that this conclusion may have been reached due to the use of a small training dataset and overfitting.
* Former methods tend to produce 1080P output, while OASIS only tests in 720P resolution. The authors need to explain the reasons. Is it limitations in data, model, computational power, or methodology that prevent scaling to higher resolutions?
* Why does the method presented in this paper consistently outperform the global pattern across all metrics in Table 3(a)? Normally, the global pattern should yield the best results. Is the model of global pattern still under-fitting, or are there other reasons?

**Questions:**

* Instead of determining the DiT head seperation by only a small set of training videos, authors could considering seperating them during inference, and develop an efficient method about it. A data-driven seperation strategy will improve its robustness.
* Authors should consider extending their methods on higher resolution like 1080P and providing corresponding video results.

---

### Official Review · Reviewer_caLv · 2025-10-29

**Soundness:** 3
**Presentation:** 3
**Contribution:** 3
**Rating:** 4
**Confidence:** 4

**Summary:**

This paper proposes OASIS, an efficient one-step diffusion model featuring attention specialization for real-world video super-resolution. OASIS introduces an attention specialization routine that reduces redundancy and enhances performance. Furthermore, the authors design a progressive strategy to enable better adaptation to complex, real-world scenarios. Experimental results demonstrate that OASIS achieves state-of-the-art performance.

**Strengths:**

1.The logic of this article is clear, and the description of the proposed method is comprehensive and easy to follow.
2. This paper propose an one-step diffusion model, OASIS, for real-world VSR. By incorporating an attention specialization routing, OASIS mitigates the redundancy of pretrained diffusion transformers when adapted to VSR, thereby reducing computational cost
and making the model better leverage diverse attention patterns for high-quality restoration.
3.The article presents extensive quantitative results that effectively demonstrate the efficacy of each module.

**Weaknesses:**

1.On Page 2, lines 074–081, the authors state that “many heads consistently behave in a localized manner across different videos and scales…”. However, this assertion lacks supporting references or visualizations, which makes it unconvincing.
2.In the provided demo, all real-world video motions are relatively slow. The authors should present additional results featuring normal or more dynamic motion to demonstrate that the showcased examples are not cherry-picked.
3.This paper reassigns some global heads to local heads to reduce unnecessary computation. It is necessary to provide a detailed comparison of computational time between the local and global head.
4.While other VSR methods report results on the AIGC dataset, this paper lacks such a comparison. It would be valuable to see how the proposed method performs on AIGC dataset.
5. How was Figure 3 generated, and how is each video represented within the figure? The authors are encouraged to provide a more detailed description of the figure’s production process as well as the meaning conveyed by Figure 3.

**Questions:**

See weaknesses.

---

### Official Review · Reviewer_tpfN · 2025-10-31

**Soundness:** 3
**Presentation:** 3
**Contribution:** 3
**Rating:** 4
**Confidence:** 3

**Summary:**

This paper introduces OASIS to tackle efficient VSR with proposed attention specialization mechanism. The attention specialization routing mechanism aims to mitigates the computational redundancy in model inference. Together with the proposed progressive training strategy, the model offers fast inference speed comparing other VSR SOTA model including SeedVR2 while maintaining superior VSR performance.

**Strengths:**

1.	The model is simple yet effective and the idea is easy to follow.
2.	The proposed attention specialization routing is efficient to reduce the computational cost of the attention operation by limiting it within a local window instead of global.
3.	The progressive learning and the training objective functions help the model achieve SOTA performance.

**Weaknesses:**

1.	The core contribution of the attention specification routing seems to be unrelated to the VSR task. Have you try the ASR in other video tasks like video generation or editing, which may also benefit from ASR ?
2.	The OASIS model seems to be based on Wan 2.1, and the DiT is trained with parameter. This may corrupt the generative capability of the original model, could LoRA training achieves better restoration (i.e., by preserving the diffusion generative prior) and faster convergence on VSR task?
3.	The window attention size is set to (3, 5, 5) in the paper, and how about other sizes? No ablation studies are provided to show how the settings are determined.
4.	Which part of the model updates the gradient from the perceptual loss? DiT or the Decoder?
5.	The visualization and analysis is insufficient. The restored results on complex texture like plant-based or text-based video are not provided and discussed. This part of the paper should be enhanced.

**Questions:**

Refer to the weaknesses. How is the attention window size determined? A detailed description on how the pixel supervision benefits DiT is needed. The contribution to the VSR is also needed to be clarified.

---

### Note · Authors · 2025-11-12

I have read and agree with the venue's withdrawal policy on behalf of myself and my co-authors.